A new aggressive xenograft model of human colon cancer using cancer-associated fibroblasts

Fernando-Macías Ester 1 2
Fernández-García Maria Teresa 3
García-Pérez Eva 4
Porrero Guerrero Belén 1
López-Arévalo Camilo 1
Rodríguez-Uría Raquel 1
Sanz-Navarro Sandra 1
Vázquez-Villa José Fernando 2
Muñíz-Salgueiro María Carmen 3
Suárez-Fernández Laura 2
http://orcid.org/0000-0003-3138-7642 Galván José A. 5
Barneo-Caragol Clara 2 6
García-Ocaña Marcos 7
de los Toyos Juan R. 2 8
http://orcid.org/0000-0001-9410-4262 Barneo-Serra Luis 1 2 9 lbarneo@uniovi.es
1 Service of Surgery, Hospital Universitario Central de Asturias , Oviedo , Spain
2 Instituto de Investigación Sanitaria del Principado de Asturias , Oviedo , Spain
3 Laboratory of Department of Molecular Histopathology in Animal Cancer Models, Oncology University Institute of the Principality of Asturias, University of Oviedo , Oviedo , Spain
4 Ophtalmologic Research Foundation , Oviedo , Spain
5 Translational Research Unit, Institute of Pathology, University of Bern , Bern , Switzerland
6 Laboratory of Medicine, Department of Clinical Biochemistry, Hospital Universitario Central de Asturias , Oviedo , Spain
7 Biotechnological and Biomedical Assays Unit, Technical-Scientific Services, University of Oviedo , Oviedo , Spain
8 Immunology Department, School of Medicine and Health Sciences, University of Oviedo , Oviedo , Spain
9 Surgery Department, School of Medicine and Health Sciences, University of Oviedo , Oviedo , Spain
Paranjape Anurag
Electronic publication date: 2020 Jun 3
Publication date: 2020
Volume: 8
Electronic Location ID: e9045
Received 2019 Nov 18; Accepted 2020 Apr 2
Copyright: © 2020 Fernando-Macías et al.
Copyright year: 2020
Copyright holder: Fernando-Macías et al.
License: This is an open access article distributed under the terms of the Creative Commons Attribution License, which permits unrestricted use, distribution, reproduction and adaptation in any medium and for any purpose provided that it is properly attributed. For attribution, the original author(s), title, publication source (PeerJ) and either DOI or URL of the article must be cited.
License URL: https://creativecommons.org/licenses/by/4.0/

Keywords: Xenograft, Animal model, CAFs, Fibroblasts culture, ProColXIα1

Funding: European Union INNPACTO-ONCOPAN IPT-010000-2010-31 Ministry of Science and Innovation, Spain by FISS-09-PS09/01911 Principality of Asturias, Spain FC-11-PC10-23 Progenika Biopharma, Spain This research has been co-financed by ERDF Funds from the EuropeanUnion; by INNPACTO-ONCOPAN IPT-010000-2010-31 Project; by FISS-09- PS09/01911 Project, Ministry of Science and Innovation, Spain; by FC-11-PC10-23 Project, FICYT, Axe 1 of the 2007-2013 ERDF Operational Framework Program of the Principality of Asturias, Spain; and by Progenika Biopharma, S.A. and Oncomatryx, S.L. Derio, Spain. The funders had no role in study design, data collection and analysis, decision to publish, or preparation of the manuscript.

==============================
Background

Colorectal cancer is the second leading cause of cancer death. Almost half of the patients present recurrence within 5 years after the treatment of the primary tumor, the majority, with metastasis. On the other hand, in the search for new animal models that simulate metastatic cancer, it has been suggested that fibroblasts immersed in the peritumoral stroma (cancer-associated fibroblasts (CAFs)), play a relevant role in the development of cancer. The objective of this study was to identify an adequate animal model to study metastatic colon cancer and the application of new treatments.

Methods

Human CAFs and normal fibroblasts (NF) for transplant and culture were obtained from surgical fresh samples of patients with adenocarcinoma of sigmoid colon. Stromal cell purity was evaluated by morphology and immunostaining with vimentin (VIM) as a fibroblast marker and anti-proColXIα1 as a specific human CAF marker. Phenotypic characterization of cultured stromal cells was performed by co-staining with mesenchymal and epithelial cell markers. For identification in mice, human CAFs were labeled with the PKH26 red fluorescence dye. Cell line HT-29 was used as tumor cells. Transplant in the head of the pancreas of 34 SCID mice was performed in four different groups, as follows: I. 150,000 CAFS (n = 12), IIa. 1.5 million HT29 cells (n = 7), IIb. 150,000 NF+1.5 million HT29 cells (n = 5), III. 150,000 CAFS+1.5 million HT29 cells (n = 10). After euthanasia performed one month later, histological analysis was made using hematoxylin–eosin and anti-proColXIα1. A histopathological score system based on three features (tumor volume, desmoplasia and number of metastasized organs) was established to compare the tumor severity.

Results

The CAFs and NF cultured were proColXIα1+/VIM+, proColXIα1/alphaSMA+ and proColXIα1+/CK19+ in different proportions without differences among them, but the CAFs growth curve was significantly larger than that of the NF (p < 0.05). No tumor developed in those animals that only received CAFs. When comparing group II (a + b) vs. group III, both groups showed 100% hepatic metastases. Median hepatic nodules, tumor burden, lung metastases and severity score were bigger in group III vs group II (a + b), although without being significant, except in the case of the median tumor volume, that was significantly higher in group III (154.8 (76.9–563.2) mm3) vs group II (46.7 (3.7–239.6) mm3), p = 0.04. A correlation was observed between the size of the tumor developed in the pancreas and the metastatic tumor burden in the liver and with the severity score.

Conclusion

Our experiments demonstrate that cultured CAFs have a higher growth than NF and that when human CAFs are associated to human tumor cells, larger tumors with liver and lung metastases are generated than if only colon cancer cells with/without NF are transplanted. This emphasizes the importance of the tumor stroma, and especially the CAFs, in the development of cancer.

Introduction

Colorectal cancer is the third most frequent cancer worldwide, with 1.8 million estimated new cases in 2018. In terms of mortality, colorectal cancer is the second leading cause of cancer death, after lung cancer, with 880,000 deaths in 2018 (Ferlay et al., 2019; Stewart & Wild, 2014). Colon cancer is three times more frequent than rectal cancer, and 97% of colon tumors are adenocarcinomas (Scientific American Surgery, 2016). Worldwide there is large geographic variability in incidence, and these differences are similar in both sexes (Ferlay et al., 2019; Sociedad Española de Oncología Médica, 2017). The highest rates of colon cancer correspond to industrialized countries (Asociación Española contra el Cáncer, 2018). Although survival rates of advanced stage tumors have improved significantly, almost 50% of patients present with cancer recurrence within 5 years following treatment of the primary tumor, with the majority of these cases having liver metastases (Rashidi et al., 2000; Roy & Majumdar, 2012).

The objective of this study was to identify an adequate animal model to study metastatic colon cancer and the application of new treatments. The ideal animal model needs to meet criteria of technical ease, a high degree of implantation in a short time period, be reproducible, and with a natural history similar to the human disease. To meet these requirements and facilitate the development of cancer, the peritumoral stroma and the fibroblasts immersed in it—cancer-associated fibroblasts (CAFs)—play a relevant role (Van Pelt et al., 2018; Mukaida & Sasaki, 2016). CAFs constitute a heterogeneous group of peritumoral fibroblasts that have properties similar to those of mesenchymal stem cells (Kalluri, 2016; Valcz et al., 2014; Huang et al., 2014; Öhlund, Elyada & Tuveson, 2014). It is thought that fibroblasts facilitate tumor growth and favor an immunosuppressive environment that avoids the elimination of tumor cells and promotes the sequestration of drugs and the radioprotection of tumor cells (Whittle & Hingorani, 2019). Therefore, the co-transplant of tumor cells with CAFs could be of interest in animal models of human cancer. The reasons that justify co-transplanting with CAFs are that they secrete growth factors, promote angiogenesis, transfer substrates to neighboring cells, and decrease the activity of natural killer cells to modulate the immune response. Previous evidence has shown that the conjoint implantation of cancerous cells with CAFs in SCID (severe combined immune deficiency) mice results in a very aggressive tumor with large local growth and liver metastases 3–4 weeks post-transplant (Porrero Guerrero, 2017). This, along with the characteristics mentioned above, led us to develop a highly aggressive heterotopic human colon adenocarcinoma xenotransplantation model, whose results we present in this article. The pancreas was selected as the implantation site as it is a site rich in growth factors that would facilitate local growth and distant growth from the implanted cells.

Materials and Methods

Patient characteristics and surgical samples

The CAFs for transplant were obtained from an 81-year-old woman with moderately differentiated adenocarcinoma of the sigmoid colon, pT3 pN0. The human fibroblasts for culture were obtained from three patients: (1) an 80-year-old male with a well-differentiated mucinous adenocarcinoma of the right colon, pT2 pN0; (2) a 72-year-old female with a moderately differentiated adenocarcinoma of the sigmoid colon, pT3 pN0; and (3) a 56-year-old male with a recurrence of moderately differentiated adenocarcinoma of the sigmoid colon, pT3pN0.

Cell cultures

Colon fibroblasts

Samples were obtained from the tumor area (for other studies), peritumoral area (for CAFs) and normal area (for “normal fibroblasts”, NF) using different surgical blades to avoid contamination few minutes after the operation. The lack of tumor cells was verified by microscopy. The samples were cut into smaller fragments. The establishment of cell cultures has been published previously (García-Pravia et al., 2013). Briefly, the fragments were enzymatically digested with collagenase Type I, the pellet was suspended in a fibroblast culture medium with fetal bovine serum (FBS) as below and antibiotics. The tissue fragments that had not been digested with collagenase underwent a second digestion with trypsin and EDTA. Cells obtained were seeded in six-well plates using fibroblast culture medium and maintained at 37 °C in a 5% CO2 incubator. All stromal cells were used at early passages (passages 3–6). The cell purity of stromal cells was assessed by morphology and by immunostaining for vimentin. Following counting in the Neubauer chamber, the cell purity of the stromal cells was evaluated for morphology, and immunostaining with VIM as a fibroblast marker, and with anti-proColXIα1 mousse mAb (DMTX1, Oncomatryx, Derio, Vizcaya, Spain) as a specific marker for human CAFs (García-Pravia et al., 2013; Galván et al., 2014; García-Ocaña et al., 2012). The remaining cells were frozen using DMEM +10% FCS and 10% DMSO as cryoprotectant in cryovials. The cryovials were maintained in a −80 °C freezer overnight and then transferred into a liquid nitrogen container for long-term storage for other studies. Both types of cells (NF and CAFs), obtained from three patients, were seeded in six-well plates at a rate of 1 × 104 viable cells per well, using the culture medium previously used. Every two days, one of the dishes was trypsinized and the cells were counted on a Neubauer chamber. The wells were not filled in any case, so there was no contact inhibition.

Colon adenocarcinoma cell line

HT-29 cells ( HTB-38; ATCC®, Manassas, VA, USA) were cultivated in DMEM standard media with 10% FBS. All the cultures were carried out in a humidified atmosphere of 5% CO2 in air at 37 °C. Culture passages and cell collections were done with trypsin/EDTA 0.05%/0.02% (Biochrom, Cambridge, UK).

Immunocytofluorescence of cultured stromal cells and confocal microscopy

The methodology used has been described previously (García-Pravia et al., 2013). In summary, cells were fixed in acetone, dried at room temperature, and then taken into the wash buffer. The samples were incubated with the anti-proColXIα1 mAb, (DMTX1, Oncomatryx, Derio, Vizcaya, Spain), Cytokeratin 19 (CK19) antibody, αSMA and VIM antibody, at room temperature, under the conditions specified in Table 1. The secondary antibodies used were green anti-mouse Alexa-488 and red anti-rabbit Alexa-546, and the sections were mounted with mounting medium containing DAPI. The colocalization was visualized and photographed using a confocal microscope with specific sources of illumination for each fluorochrome excitation.

Table 1 Summary of antibodies used for immunocytochemistry (IHC).

Primary antibodies (species)	Clone	Commercial reference	Dilution	Incubation time (min)	
anti-procollagenXIα1 (mAb)	1E8.33	DMTX1/Oncomatryx, Spain	1:400	30	
CK 19 (mAb)	RCK108	Dako, Denmark	1:50	15	
α-SMA (mAb)	1A4	Dako, Denmark	Ready to use	20	
Vimentin (pAb)	C-20	Santa Cruz Biotech, Germany	1:400	10	
Note:

mAb, Monoclonal antibody; pAb, Polyclonal antibody.

Identification of human CAFs inside the pancreas of mice

CAFs were labeled with PKH26 red fluorescence cell dye (Sigma-Aldrich PKH26 Red Fluorescent Cell Linker Kits for General Cell Membrane Labeling). The labeling vehicle provided in the kits (Diluent C) was designed to maintain cell viability, while maximizing dye solubility and staining efficiency during the labeling step. Due to it, cells present extremely stable fluorescence and PKH26 is the cell linker dye of choice for in vivo cell tracking studies. PKH26 consists of an aliphatic molecule that is incorporated into the cell membrane, linked to a rhodamine-like fluorescent dye. The optimum excitation wavelength is 551 nm and the emission wavelength is 567 nm. The CAFs were injected into the pancreas of mice. After seven days, mice were sacrificed and tumor samples were embedded in Tissue-Teck OCT (Thermo, Waltham, MA, USA), and frozen in liquid nitrogen. Five-micrometer-thick sections were stored at −80 °C. Sections were fixed in acetone (−20 °C, 10 min) and incubated with anti-proColXIα1 mAb (Oncomatryx, Derio, Vizcaya, Spain, 1:400, 30 min) secondary conjugated Alexa 488 (1:500, Invitrogen, Carlsbad, CA, USA). Nuclei were counterstained with 4,6-diamidino-2-phenylindole dihydrochloride hydrate (DAPI, Vector Labs, Burlingame, CA, USA). To document the presence and human origin of CAFs in tumor sections of mice, various sections per mouse were analyzed by confocal microscopy. PKH26 positive cells presented red fluorescence and pro-ColXIα1 positive cells presented green fluorescence.

Animals

The transplants were done in immunodeficient male mice, BALB/cJHanHsd-Prdkcscid SCID strain (Envigo RMS, Spain SL), aged 4–10 weeks. They remained housed in a special room for immunodepressed mice, on a ventilated rack, and under a laminar flow hood used for handling.

Transplants

Inhalational anesthesia (isoflurane) and preoperative analgesia (intraperitoneal buprenorphine 0.05 mg/kg 15 min prior to the intervention) were used. A 1–2 cm laparotomy was performed and the cell solution was slowly injected (30 s via a 30 G needle, 50 µl in PBS) into the head of the pancreas at the level of the pylorus. Following this procedure, meloxicam 1.5–2 mg/kg was administered intraperitoneally. The buprenorphine and meloxicam doses were repeated every 24 h during two days via the subcutaneous route.

Experimental design of transplants

Heterotopic transplant in the head of the pancreas of SCID mice

A total of 34 mice were randomly distributed in four groups: I. Twelve animals received 150,000 CAFs; IIa. In seven mice 1.5 million HT29 cells were implanted; IIb. In five animals 150,000 NF and 1.5 million HT29 cells were transplanted; III. Ten mice received 150,000 CAFs and 1.5 million HT29 cells. To make sure that the CAFs transplants would not produce tumors on their own, a greater number of animals were included in the group I. With the aim of removing any potential defects in fibroblasts isolation, aliquots of the same isolated fibroblasts were concurrently transplanted in all of the experimental groups. All processes were performed under sterile conditions.

Euthanasia

At one month post-transplant, the mice were euthanized using a CO2 chamber, and the intestinal and thoracic content removed in block, preserving them in 4% paraformaldehyde.

Histological analysis

The samples were analyzed by personnel of the Department of Molecular Histopathology in Animal Cancer Models (University Institute of Oncology). Specimens were fixed for 24 h in 4% paraformaldehyde at room temperature, embedded in paraffin, sectioned in sections of 5 µm thickness, and stained with hematoxylin–eosin and anti-proColXIα1 mAb (DMTX1; Oncomatryx, Derio, Vizcaya, Spain). The volume of the tumor which developed in the pancreas was calculated in mm3 with the formula major axis x (minor axis)2 x 0.53. To assess the size of liver metastases, each liver was completely cut into serial sections with a distance between each section of 200 µm. The liver metastases were evaluated by counting the number that were observed in the histological sections, classifying their size in small, medium and large, giving to these a value of 1, 2 or 3, respectively, as a measure of liver tumor burden. A small metastasis size was considered when the same tumor appeared in a single section (≤400 µm), medium when the tumor appeared in two consecutive sections (>400 μm and ≤600 µm), and large when the tumor appeared in three or more sections (>600 µm). Hence, an animal with a liver containing one small metastasis, one medium and one large metastasis, was classified with a tumor burden of 6 (1 × 1 + 1 × 2 + 1 × 3 = 6).

We established a histopathological score system from 3 to 9 points, to compare the tumor severity or aggressiveness, based on three histopathological features (tumor volume, desmoplasia and number of metastasized organs) according to the following score: tumor volume feature was:1 = small (<50 mm3); 2 = medium (50–150 mm3) and 3 = large (>150 mm3); the score given for desmoplasia was: 1 = mild; 2 = mild/moderate; 3 = moderate/severe; the score given for number of metastasized organs was: 1 = 1–2 organs; 2 = 3–4 organs; 3 = >4 organs.

Statistical analysis

Quantitative variables were expressed as medians. Groups were compared using the Mann–Whitney U test. The association between quantitative variables was assessed using the regression coefficient and the prediction of one variable from another via the regression equation. The growth of the cell cultures was analyzed using ANOVA with Bonferroni correction. All analyses were performed using the programs SPSS 15.0 (SPSS, Inc., Chicago, IL, USA) for windows and MedCalvs12.

Ethical considerations

All experiments complied with the European Union (2010/63/UE) and Spanish (RD 53/2013; ECC/556/2015) standards, and were in accordance with the guidelines of the Committee for the handling and care of animals of the University of Oviedo. The Committee for the handling and care of animals of the University of Oviedo provided full approval for this research (PROAE 01/2016). The patients signed consent forms indicating their willingness to participate in the study.

The extraction of surgical samples was approved by the Hospital Universitario Central de Asturias ethical committee (Project no 42/12).

Results

Characterization of the colon adenocarcinoma CAFs

The fibroblasts for transplantation had positive immunostaining to proColXIα1 (Fig. S1). Using confocal microscopy, the CAFs and NF cultured were proColXIα1+/VIM+, proColXIα1+/alphaSMA+ and a small number of cells with the epithelial phenotype (proColXIα1+/CK19+) (Fig. 1). Figure S2 shows the coexpression in the peritumoral area of carcinoma in mouse heterotopic xenogratfs of the PKH-26 dye and human proColXIα1 in CAFs. The CAFs growth curve was significantly larger than that of the NF (p < 0.05) (Fig. 2). The details of data collected in growth curves are depicted in Table S1 in which the number of cells obtained in each well is represented as the mean of two determinations.

Figure 1 Confocal microscopy of cultured CAFs.

Double fluorecence stain illustrates the presence of: (A) Cell proCOL11A1+/VIM+, (B) ProCOL11A1+/CK19+ and (C) ProCOL11A1+/alphaSMA+. Red, proCOL11A1; green, VIM, alphaSMA and CK19; blue, nuclei. Scale bar: (A and B) 20 µm, (C) 100 µm (X630).

Figure 2 Growth curves of fibroblasts.

Blue, normal fibroblasts; green, CAFs. Cultures obtained from three patients with adenocarcinoma of colon. Mean ± 2 SEM of three patients with duplicate determinations.

Xenotransplants

No tumor developed in those animals that only received CAFs (Table 2). Of the animals that received HT29 cells without fibroblasts, five died in the initial days following the procedure. The surviving two mice presented with poorly differentiated carcinomas in the pancreas with light dysplasia, infiltrating the stomach due to proximity, and liver metastases; lung metastases also developed in one of the mice in which a focus was also found in the small intestine (Fig. 3; Table S2). Of the five mice that received HT29 + NF, one died; the remaining four developed a tumor with liver metastases, and one had multiple lung and spleen metastases (Fig. 4; Table S3). They all showed occasional tumor foci in the small and large bowel.

Table 2 Summary of results in xenografts.

	I. CAFs	II*. (HT29) + (HT29 + NF)	III. HT29 + CAFs	p-value II vs III	
No. animals	12	12 (7 + 5)	10		
Exitus	0	6 (5 + 1)	2		
Tumor incidence (%)	0	100 (6/6)	87.5 (7/8)	NS	
Tumor volume (mm3)	–	3.7; 91.6 8; 28.2; 65.3; 239.6	76.9; 93.7; 143.9; 154.8; 169.9; 207; 563.6		
Median tumor volume (mm3) (min–max)	–	46.7 (3.7–239.6)	154.8 (76.9-563.2)	0.04	
Hepatic metastases (%)	–	100 (6/6)	100 (7/7)		
No. hepatic nodules		1; 6 2; 3; 4; 5	6; 9; 9; 10; 14; 14		
Median hepatic nodules (min–max)		3.5 (1–6)	6 (2–11)	0.12	
Median tumor burden (min–max)		6.5 (1–12)	10 (6-15)	0.08	
Lung metastases (%)	–	33 (2/6)	57 (4/7)	NS	
Severity score (min–max)		5 (3–8)	7 (5–8)	0.06	
Note:

CAFS, cancer-associated fibroblasts; NF, normal fibroblasts. *Group II: IIa (HT29) + IIb (HT29 + NF). See explanation in Study Limitations chapter. Numbers in bold italics correspond to group IIb. Liver tumor burden and severity score: see text.

Figure 3 HT-29 in the pancreas model.

(A) Microphotograph of maximal tumor size in pancreatic tissue (maximal tumor size, score 1) 1.25×, H–E. Red circle, maximal tumor size in pancreatic tissue. (B) Histological image of poorly differentiated pancreatic adenocarcinoma, 10×, H–E. Malignant epithelial cells isolated or arranged in small or large clusters, with mild-moderate desmoplastic stromal reaction (score 2). (C) The image shows metastasis of a small cluster of malignant epithelial cells (score 1 of no. of metastasized organs) in: C, liver tissue; D, lung tissue (H–E, 4×). Red circle (C and D) nest of tumoral cells in liver and lung, respectively. Scale bar: (A) 500 µm; (B and D) 50 µm; (C) 200 µm.

Figure 4 HT29 + NF.

(A) Mouse necropsy. Great pancreatic tumor distention gallbladder (Courvoisier–Terrier signe), gastric infiltration. (B) Microphotograph of maximal tumor size in pancreatic tissue (maximal tumor size, score 2) 1.25×, H–E. (C) The image shows metastasis of small or medium clusters of malignant epithelial cells (score 1 of no. of metastasized organs) in: C, liver tissue; D, lung tissue (H–E, 4×). Scale bar: (B) 500 µm; (C) 200 µm; (D) 50 µm.

Of the ten animals in the HT29 + CAFs group, two died and one did not develop tumor. The tumor volume in the pancreas was significantly bigger than in the combined group that received HT29 with/without NF. All cases with tumor in the pancreas developed liver metastases with a higher tumor burden than in the other transplant groups, but without reaching statistical significance, and in more than half lung metastases were identified. One animal presented with metastases in the spleen, four in the small intestine, and two in the peritoneum with multiple foci of lymphovascular invasion (Fig. 5; Fig. S3; Table S4).

Figure 5 HT29 + CAFs.

(A) Mouse necropsy. Great pancreatic tumor with liver metastases, distention gallbladder (Courvoisier–Terrier signe), gastric infiltration. (B) Microphotograph of maximal tumor size in pancreatic tissue (maximal tumor size, score 3) 1.25×, H–E. Red circle, maximal tumor size in pancreatic tissue. (C) Histological image of poorly differentiated pancreatic adenocarcinoma, 10×, H–E. Malignant epithelial cells isolated or arranged in small or large clusters, with necrosis and moderate desmoplastic stromal reaction (score 2). Scale bar: (B) 500 µm; (C) 50 µm.

The cytological pattern of all of these showed cell and nuclear pleomorphism, with an increase in the nuclear–cytoplasmic ratio, irregular nuclear membranes, and various nucleoli. The histological pattern of the mice that received fibroblasts or CAFs was that of moderately or poorly differentiated carcinoma, with low–moderate dysplasia, some with vascular invasion, and infiltrating the stomach due to proximity.

Median hepatic nodules, tumor burden, lung metastases and severity score were bigger in group III vs group II, although without being significant, except in the case of the median tumor volume, that was significantly higher in group III (154.8 (76.9–563.2) mm3) vs group II (46.7 (3.7–239.6) mm3), p = 0.04. A relationship was observed between the size of the tumor developed in the pancreas and the metastatic tumor burden in the liver (y = 0.321 + 4.483 log(x), r = 0.69, p = 0.001) and the severity score (y = 2.231 + 2.008 log(x), r = 0.79, p < 0.001) (Fig. 6).

Figure 6 Relationship between pancreatic tumor and hepatic burden (A) or severity score (B).

Regression ecuations. (A) Pancreatic tumor size-hepatic burden and (B) pancreatic tumor size-severity score.

Discussion

The possibility of personalized cancer treatment is affected by the heterogeneity of the primary tumor and that of the metastases, which means that the results of gene and molecular expression obtained from one area of the tumor are different from those in other areas of the same tumor, and therefore the behavior cannot be determined from a few biopsies (Gerlinger et al., 2012). In an attempt to simulate the human disease in animals and in order to be able to apply new treatments, different animal models of cancer have been made (Johnson & Fleet, 2013; Taketo & Edelmann, 2009). Animals that have been genetically modified for this purpose, GEMM, may shed light on the role of different genes in the appearance of this disease. In one of these animal models, it was demonstrated that cancer metastases precede the formation of the primary tumor (Rhim et al., 2012), which would be in line with the hypothesis that cancer is a systemic disease from the start; this is further supported by findings of metastases in patients without being able to localize the primary tumor. The main disadvantage of these models is that the alterations affect the germ line, resulting in the possible development of diseases from the embryonic stage. Furthermore, they develop precancerous or cancerous lesions at other levels (Johnson & Fleet, 2013).

The initial control of the tumor is an advantage in chemically induced cancers or cancers induced via implantation of tumor cells (Johnson & Fleet, 2013; Karim & Huso, 2013). In terms of the administration of carcinogenic substances, the process depends on the duration and dose, and the state of the organism (Karim & Huso, 2013). One general limitation of these experiments is the scarce development of metastases (Johnson & Fleet, 2013).

The model most utilized by researchers to trial new treatments is the xenotransplant of cancerous human cells in immunodepressed animals, usually SCID mice or other variants, either in the organ where the tumor develops—orthotopic xenotransplant—or in another anatomic region such as subcutaneous tissue—heterotopic xenotransplant. An important limitation of this model is that it lacks the influence that part of the immune system has on the development of the xenotransplant. In xenotransplants, tumor cell lines or slices of tumor samples extracted from patients, patient derived xenograft (PDX), are used. These samples are obtained fresh from tumor and should be implanted as soon as possible within the first 24 h (Rashidi et al., 2000; Fu et al., 1991). PDX has the advantage that the animal receives the full tumor repertoire from the patient: tumor and the peritumoral inflammatory reaction (Katsiampoura et al., 2017). Additionally, these would be appropriate to trial personalized treatments: the patient’s tumor is transplanted in a series of animals that are treated with diverse drugs, and the one which best controls the tumor would be used in the patient (Okada, Vaeteewoottacharn & Kariya, 2018; Williams, 2018). Nevertheless, this method has its limitations as well: it requires the use of animals that lack an immune system, the stroma that forms is murine and not human, and sometimes does not develop metastases (Katsiampoura et al., 2017). In addition, its clinical use is not without difficulties: with slow tumor growth, it has to be transferred to other animals to have a sufficient number for treatments, need for controls and costs involved.

One of the primary surgical models developed was the heterotopic transplant conducted in subcutaneous tissue of immunodeficient mice and implanted with human colon cancer cells. In this model, the growth was easily monitored, however, being situated extra-abdominally, did not develop metastases (Mittal et al., 2015). Some authors have utilized this technique as a step prior to conducting an orthotopic transplant, using the subcutaneous tumor to obtain samples that are then implanted in the colon of other mice (Mittal et al., 2015; Flatmark et al., 2004; Fodstad, 1991). The splenic capsule and kidney are also used as sites for implantation (Mittal et al., 2015). The orthotopic model is considered the one that best simulates the process of human colon cancer development (Mittal et al., 2015).

In recent years organoids have been developed, which are bodies of cells cultivated in vitro forming a three-dimensional structure. Their functional and anatomic characteristics are more like the original organ. They are made from healthy and tumor stem cells, which allows for the study of its behavior and reaction to drugs in both cases (Young & Reed, 2016).

To avoid the inconvenience of the scarce tumor appearance, slow development, absence of metastases and economic cost, that the aforementioned models present, we have trialed a new xenotransplant model that consists of implanting CAFs obtained from fresh colon cancer specimens conjointly with HT-29 tumor cells in SCID mice. This model is very aggressive, and within a few weeks develops a large tumor with liver metastases and therefore it is possible to start new treatments without delay. Our team has focused its research on CAFs (García-Pravia et al., 2013; Galván et al., 2014; García-Ocaña et al., 2012; Fuentes Martínez et al., 2015), and on a protein from the collagen family, collagenXIα1, which is universally overexpressed in the stroma of epithelial tumors (Vázquez-Villa et al., 2015), namely in the cytoplasm of the CAFs and not in the epithelial cells. The antibody anti-proColXIα1 stains CAFs specifically (García-Ocaña et al., 2012), in contrast to other mesenchymal markers such as VIM and αSMA that stains a wide variety of fibroblasts, including NF, fibroblasts from inflammatory areas, and in lesions, etc. In tissue, whereas non-peritumoral fibroblasts are stained with anti-VIM and anti αSMA, the CAFs are also stained with anti-proColXIα1; nevertheless, in cultures these differences in the expression of markers between the different groups disappear. In a pancreatic cancer model, cultivating fibroblasts from CAPAN-1 cells, a higher proliferation of these cells was observed in the presence of peritumoral fibroblasts compared with non-tumor fibroblasts or without the presence of fibroblasts (Porrero Guerrero, 2017); which would be consistent with the results of the xenotransplants associated with the CAFs. The human fibroblasts transplanted in animals can be visualized using anti-proColXIα1 and be distinguished from those of the animals, as this antibody is specifically human and does not cross-react with other species (Fernández-García et al., 2014), which would allow the monitoring over time of these cells in the animal. Another feature of anti-proColXIα1 is that it marks cells that are in epithelial–mesenchymal transition showing double staining for epithelial (CK19) and mesenchymal (VIM) (García-Pravia et al., 2013).

The resulting cultured cells in our experiments were fibroblasts both morphologically and in terms of their phenotypic characteristics, staining for nonspecific mesenchymal markers such as VIM; specific markers of inflammation, such as smooth muscle alpha-actin; and, as well as the specific marker for CAFs, procollagenXIα1. The percentage of mesenchymal cells marked exclusively with CK19 was scarce, although it was possible to observe cells with a double phenotype procollagenXIα1+/CK19+, characteristic of cells in epithelial–mesenchymal transition. As our samples of CAFs were not significantly contaminated with tumor cells, this shows why the transplants conducted with only fibroblasts did not develop any tumor. In other models of xenotransplants, tumor did not develop either with the administration of CAFs (Porrero Guerrero et al., 2015; Rodríguez Uría et al., 2018). The fibroblasts of normal tissue did not stain for SMA nor for procollagenXIα1; however, under the artificial conditions of the culture, some fibroblasts obtained from non-tumor areas acquired positivity for those markers due to the artificial conditions to which they had been subjected.

According to the hypothesis that the tumor stroma and fibroblasts in particular, promote tumor progression, we coupled CAFs cultivated from our surgical samples with HT29 cells in this new transplant model and chose the pancreas as the implantation site, as in this organ very aggressive tumors develop with a high metastatic capacity and high mortality. This is due to various mechanisms, among which are the influence of the substances secreted by the pancreatic islets that stimulate cell growth (Kim & Herbrok, 2001). In our research group, models of pancreatic cancer have been developed previously (Porrero Guerrero, 2017). Given our training in this technique we decided to apply the heterotopic model of colon cancer. We conducted a control group with CAFs transplant without tumor cells to verify that these cells alone are not capable of developing tumor. In the subsequent experimental groups, cell lines of colon cancer were transplanted, alone and coupled with NF or CAFs. In both models, tumors and multiple metastases were obtained, with the group including CAFs producing the largest size tumors.

Conclusions

In conclusion, our experiments demonstrate that the association of human peritumoral fibroblasts with human tumor cell generates, early, larger tumors with liver and lung metastases, than if only colon cancer cells are transplanted. This demonstrates the importance of the tumor stroma and especially the CAFs, in the development of colon cancer.

Study limitations

The loss of animals in the HT29 group made it impossible to compare them directly with the animals that received NF and CAFs, although in other experiments it was found that tumor development and metastases were lower than when associated with CAFs (Porrero Guerrero et al., 2013, 2015; Rodríguez Uría et al., 2018; ); hence, we decided to regroup the HT29 and HT29 + NF animals in one group, assuming that the NF would not influence tumor growth as much as the CAFs.

Supplemental Information

Supplemental Information 1 Cultured CAFs proCOL11A1+ before their transplantation.

Scale bar: 20 µm.

Click here for additional data file.

Supplemental Information 2 Coexpression of PKH-26 (red) and human pro-ColXIα1 (green) in CAFs (peritumoral area of carcinoma in mouse heterotopic xenografts).

Sections were counterstained with DAPI (blue), nuclear cell marker. Original magnification, ×630. Scale bar: 20 µm.

Click here for additional data file.

Supplemental Information 3 HT-29+CAFS.

Metastasis of small and large clusters of malignant epithelial cells with desmoplastic reaction in some of them (score 3 of no. of metastasized organs) in: (A) Skin; (B) Spleen; (C) Liver; (D) Lung. (H–E, 4×). Scale bar: (A, B and C) 200 µm; (D) 50 µm.

Click here for additional data file.

Supplemental Information 4 Growth of fibroblast cultures.

Twelve-day growth data of normal human fibroblasts (NF) and cancer-associated fibroblasts (CAF), obtained from three patients with sigmoid colon adenocarcinoma (identification code: 34, 40 and 49).

Click here for additional data file.

Supplemental Information 5 Report of HT29 transplantation in the head of the pancreas of SCID mice.

Histopathological report of HT29 transplantation in the head of the pancreas of SCID mice.

Click here for additional data file.

Supplemental Information 6 Report of HT29 + NF transplantation.

Histopathological report of HT29 + NF transplantation in the head of the pancreas of SCID mice.

Click here for additional data file.

Supplemental Information 7 Report of HT29 + CAFs transplantation.

Histopathological report of HT29 + CAFs transplantation in the head of the pancreas of SCID mice.

Click here for additional data file.

Supplemental Information 8 Raw data.

Cultures time and tumor characteristics.

Click here for additional data file.

Additional Information and Declarations

Competing Interests

Author Contributions

Human Ethics

Animal Ethics

Patent Disclosures

Data Availability

The proCOL11A1 mAb has been patented by Oncomatryx, S.L (PCT/ES2012/070616; WO 2013/021088 A2).

Ester Fernando-Macías performed the experiments, prepared figures and/or tables, authored or reviewed drafts of the paper, and approved the final draft.

Maria Teresa Fernández-García analyzed the data, prepared figures and/or tables, authored or reviewed drafts of the paper, and approved the final draft.

Eva García-Pérez performed the experiments, prepared figures and/or tables, and approved the final draft.

Belén Porrero Guerrero performed the experiments, authored or reviewed drafts of the paper, and approved the final draft.

Camilo López-Arévalo performed the experiments, authored or reviewed drafts of the paper, and approved the final draft.

Raquel Rodríguez-Uría performed the experiments, authored or reviewed drafts of the paper, and approved the final draft.

Sandra Sanz-Navarro performed the experiments, authored or reviewed drafts of the paper, and approved the final draft.

José Fernando Vázquez-Villa performed the experiments, analyzed the data, authored or reviewed drafts of the paper, cultures of fibroblasts. Generation of mAb anti-proColXIα1, and approved the final draft.

María Carmen Muñíz-Salgueiro analyzed the data, prepared figures and/or tables, and approved the final draft.

Laura Suárez-Fernández performed the experiments, prepared figures and/or tables, and approved the final draft.

José A. Galván analyzed the data, prepared figures and/or tables, and approved the final draft.

Clara Barneo-Caragol analyzed the data, authored or reviewed drafts of the paper, and approved the final draft.

Marcos García-Ocaña performed the experiments, prepared figures and/or tables, cultures of fibroblasts. Generation of mAb anti-proColXIα1, and approved the final draft.

Juan R. de los Toyos performed the experiments, prepared figures and/or tables, authored or reviewed drafts of the paper, cultures of fibroblasts. Generation of mAb anti-proColXIα1, and approved the final draft.

Luis Barneo-Serra conceived and designed the experiments, analyzed the data, prepared figures and/or tables, authored or reviewed drafts of the paper, and approved the final draft.

The following information was supplied relating to ethical approvals (i.e., approving body and any reference numbers):

The extraction of surgical samples was approved by the Hospital Universitario Central de Asturias ethical committee (Project no 42/12).

The following information was supplied relating to ethical approvals (i.e., approving body and any reference numbers):

All experiments complied with the European Union (2010/63/UE) and Spanish (RD 53/2013; ECC/556/2015) standards, and were in accordance with the guidelines of the Committee for the handling and care of animals of the University of Oviedo. The Committee for the handling and care of animals of the University of Oviedo provided full approval for this research (PROAE 01/2016).

The following patent dependencies were disclosed by the authors:

The antibody detailed in this study is under a patent filed by Drs. Luis Barneo, Juan R. De Los Toyos, Marcos García-Ocaña and others titled: METHODS AND PRODUCTS FOR IN VITRO DIAGNOSIS, IN VITRO PROGNOSIS AND THE DEVELOPMENT OF DRUGS AGAINST INVASIVE CARCINOMAS (PCT/ES2012/070616; WO 2013/021088 A2). There are no other patents, products in development or marketed products to declare.

The following information was supplied regarding data availability:

The raw measurements are available in the Supplemental File.

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
