# Peer review of "A new aggressive xenograft model of human colon cancer using cancer-associated fibroblasts"

_PeerJ, doi:10.7717/peerj.9045_

## Round 0.1 · original submission · Major Revisions

The reviewers found your work as useful to the scientific community. However, they have raised several concerns including missing labels, controls, misspellings, and missing details. Address each of the concerns of both reviewers including additional experiments.

Reviewer 1 ·

Basic reporting

Methods section should be written in preterit (line 143,144). αSMO in line 287, did the authors mean αSMA? In line 156 replace the comma with a point.
References 6 and 24 are the same (Rashidi et al., Anticancer Res 2000; 20:715-22). In terms of CAF, one of the most comprehensive reviews is missing (Kalluri, Nat Rev Cancer 2016; 16:582-598).
Figure 1: In the figure legend αSMA is missing, for green only VIM and CK19 are indicated. In the legend the scale bars are indicated with 30 µm but under the bars within the pictures is written 20 µm. An explanation about the insert in 1C is missing.
Figure 2: The label and numbering of the Y-axis are partially overlaid. Depending on final figure size the axis numbering might be too small. Moreover, statistical analysis of standard deviation would gain more relevance than the indicated SEM.
Figure 4: misspelling in figure legend: HT29 + FN should be HT29 + NF; also in Table 2 head of column
Figure 3, 4, 5, S1, S2, S3: No or too small scale bars are provided. The specification of magnification is not sufficient, for example after enlarging an image section the cells appear larger but the magnification stays the same.

Experimental design

The authors confirmed in the online form that for animal handling the name of the approval organization and approval number appear in the Methods section, but it does not.
αSMA is missing in methods section of immunofluorescence.
In Table 1: The antibodies were used for immunofluorescence and not for immunohistochemistry as stated. What is meant with dilution 11:00:00 for Vimentin antibody?
The authors should provide details how they performed growth kinetics, i.e. were the counted cell replated or were samples for each time point plated initially? Might the decrease in cell number for CAF be a result of overgrowth/contact inhibition?

Validity of the findings

In the results section (line 193) the authors state that cells had positive immunofluorescence to vimentin and Col11A1, but vimentin staining is not shown in Fig S1. How was the staining performed? It does not look like immunofluorescence as described in the methods section.
Did the authors performed negative controls esp. for CK19 staining to exclude unspecific staining? If there are differences in SMA and Col11A1 expression between normal fibroblasts and CAFs as stated in the discussion (line 310) the authors should provide resp. data.

Additional comments

The authors show that CAFs support tumor growth of HT29 colon cancer cells more than normal fibroblast. They show that tumor cell injection in pancreas head results in tumor growth and metastasis with the finding that the larger the tumor the more metastases were observed. These are interesting findings, but in the discussion section relevant literature regarding these findings is missing.
That CAFs support tumor and xenograft growth has been widely described (Olumi et al., Cancer Res 1999;59:5002–5011; Orimo et al., Cell 2005;121:335–348; Kalluri et al., Cancer 2006;6:392-401; Erez et al., Cancer Cell 2010;17:135-147; Hammer et al., Neoplasia 2017;19:496-508; and others).
Also the injection in pancreas for tumor growth also of small injected cell numbers was already published (Partecke et al. Eur Surg Res 2011;47:98–107; Sato et al. Int. J. Mol. Sci. 2017;18:1678).
It would be interesting using a tumor cell line that shows less metastasis than HT29, as this cell line regularly develops metastases even if subcutaneously injected with correlation of tumor burden and metastases load (Jojovic, Cancer Letters 2000;152:151-156; own observation), or to compare different cell lines.
Furthermore it would be very interesting to detect human stromal cells in tumors using human specific antibody to look whether there are differences in the amount of integrated human stromal cells in tumor stroma in dependence of injected cells (NF vs. CAF).

Reviewer 2 ·

Basic reporting

The overall text is written in proper English, albeit there are some mislabeling/spelling errors that need to be corrected.

Spelling error in M&M: “Oncomatrix” should be “Oncomatryx”

Experimental design

Author should describe the methods used for assessing growth of fibroblasts.
-Figure 3: No indication in the legend of what the circled region in panels (a), (c) and (d) represents.
-Figure 4: Mislabeling/spelling error in the title should be “NF” instead of “FN”.
-M&M: Authors mentioned using the proColXIa1 to stain the tissues. However, there was not mentioned of the staining pattern in the mouse tumour to indicate the presence/absence of transplanted human CAFs/NFs.

Validity of the findings

Overall it is an interesting model but some of the concerns regarding source of CAF, nature of transplantation needs to be addressed.

Additional comments

In this study, the primary aim was to describe a new animal model to study metastatic colon cancer, which may eventually be used for evaluation of new therapies. Unlike conventional PDX, the authors demonstrated that the co-transplantation of cancer-associated fibroblasts (CAFs) with human colon cancer line (HT-29) lead to an increase in hepatic nodules, tumour burden, lung metastases and severity score. This is in comparison to mice that had tumour alone or tumour with normal fibroblasts (NFs). In addition, they evaluated the difference between their patient derived CAFs and NFs; showing that CAFs have a higher growth and NFs. Overall, the authors discuss on a new area of cancer research, which is the study of how the tumour microenvironment affects cancer and metastasis development. It is in agreement with the authors that the use of conventional PDX strategies to date may be limited due to the lack of stromal and immune factors. Here the authors attempted to address this by showcasing their efforts to develop a metastatic mouse model with the addition of CAFs.
Authors should provide comments on following points:

1. Authors derived CAFs from the peritumoural area instead of the core tumour. Given the heterogenity of CAFs and hypoxic tumor core one wonders if CAFs from tumor core and peritumoural area are functionally similar.

2. The authors transplanted their tumour (+/- fibroblasts) in the pancreas. How this method is similar or diffrent to other well-known established methods such as orthotropic models, experimental liver metastasis (intra-splenic) or experimental lung metastasis (intravenous). Does this method recaptures the natural tumour development?

3. How did the authors decide on the ratio of fibroblasts to tumour cells? Is this from literature/ understanding of the ratio of stromal to tumour in colon cancer?

---

## Round 0.2 · accepted · Accept

Congratulations, your manuscript has been accepted for publication. As both the reviewers had pointed out numerous instances that required corrections, Meanwhile I strongly advise you to go through the manuscript again carefully as some of the errors may still have escaped. Some technical omissions may not get picked up during editing.

Reviewer 2 ·

Basic reporting

Pass

Experimental design

Pass

Validity of the findings

Pass

Additional comments

Authors have address all the questions thoughtfully. I recommend the acceptance of manuscript.